# *Salvia rosmarinus* Spenn. (Lamiaceae) Hydroalcoholic Extract: Phytochemical Analysis, Antioxidant Activity and In Vitro Evaluation of Fatty Acid Accumulation

**DOI:** 10.3390/plants12183306

**Published:** 2023-09-18

**Authors:** Vincenzo Musolino, Roberta Macrì, Antonio Cardamone, Luigi Tucci, Maria Serra, Carmine Lupia, Samantha Maurotti, Rosario Mare, Saverio Nucera, Lorenza Guarnieri, Mariangela Marrelli, Anna Rita Coppoletta, Cristina Carresi, Micaela Gliozzi, Vincenzo Mollace

**Affiliations:** 1Laboratory of Pharmaceutical Biology, Department of Health Sciences, Institute of Research for Food Safety and Health IRC-FSH, University “Magna Græcia” of Catanzaro, 88100 Catanzaro, Italy; 2Department of Health Sciences, Institute of Research for Food Safety and Health IRC-FSH, University “Magna Græcia” of Catanzaro, 88100 Catanzaro, Italy; robertamacri85@gmail.com (R.M.); studiolupiacarmine@libero.it (C.L.); saverio.nucera@hotmail.it (S.N.); annarita.coppoletta1@gmail.com (A.R.C.); gliozzi@unicz.it (M.G.); mollace@unicz.it (V.M.); 3H&AD Srl, 89032 Bianco, Italy; l.tucci@head-sa.com; 4Department of Clinical and Experimental Medicine, University “Magna Græcia” of Catanzaro, 88100 Catanzaro, Italy; smaurotti@unicz.it (S.M.); mare@unicz.it (R.M.); 5Department of Pharmacy, Health and Nutritional Sciences, University of Calabria, 87036 Rende, Italy; mariangela.marrelli@unical.it; 6Veterinary Pharmacology Laboratory, Department of Health Sciences, Institute of Research for Food Safety and Health IRC-FSH, University “Magna Græcia” of Catanzaro, 88100 Catanzaro, Italy; carresi@unicz.it

**Keywords:** Lamiaceae, *Salvia rosmarinus* Spenn., hydroalcoholic extract, polyphenolic compounds, terpenoids, rosmarinic acid, carnosic acid, carnosol, electron paramagnetic resonance (EPR), polystyrene resin, non-alcoholic fatty liver disease/metabolic-associated fatty liver disease (NAFLD/MAFLD), antioxidant activity

## Abstract

*Salvia rosmarinus* Spenn. is a native Mediterranean shrub belonging to the Lamiaceae family and is well-known as a flavoring and spicing agent. In addition to its classical use, it has drawn attention because its biological activities, due particularly to the presence of polyphenols, including carnosic acid and rosmarinic acid, and phenolic diterpenes as carnosol. In this study, the aerial part of rosemary was extracted with a hydroalcoholic solution through maceration, followed by ultrasound sonication, to obtain a terpenoids-rich *Salvia rosmarinus* extract (TR*Sr*E) and a polyphenols-rich *Salvia rosmarinus* extract (PR*Sr*E). After phytochemical characterization, both extracts were investigated for their antioxidant activity through a classical assay and with electron paramagnetic resonance (EPR) for their DPPH and hydroxyl radicals scavenging. Finally, their potential beneficial effects to reduce lipid accumulation in an in vitro model of NAFLD were evaluated.

## 1. Introduction

*Salvia rosmarinus* Spenn., belonging to the Lamiaceae family, is one of the oldest native Mediterranean shrubs. It has a powerful aroma, dark green elongated leaves and whitish, bluish- or bluish-purple flowers [1,2,3]. The name generally used, *Rosmarinus officinalis* L., is a synonym of the actual name, *Salvia rosmarinus* Spenn., because recent evidence has shown that *Rosmarinus* L. are nested in *Salvia* L. [4].

*S. rosmarinus* is a well-known aromatic and ornamental plant, and the oil or crude extract from its aerial parts has been used traditionally for several purposes [2,5]. Due to its protective effect against oxidative decay and reduction in lipid oxidation, *S. rosmarinus* was widely employed as a spice in cooking, and as a natural preservative in the food industry [6,7]. Traditionally, in the flagellation rituals of a small village situated in Calabria (Nocera Terinese, southern Italy), the flagellants called “vattienti”, after striking up their legs, used to wash the wounds with a rosemary and vinegar decoction, thanks to its soothing and disinfectant properties [8].

In addition to its use as a flavoring and spicing agent or as ingredients in cosmetics, perfumes, and lotions [9], *S. rosmarinus* has been traditionally used for medicinal purposes. Its carminative action, particularly in dyspepsia with improvements in hepatic and biliary function, has been described [10]. Its digestive, diuretic, balsamic [11] as well as rubefacient [12] properties have been extensively exploited in traditional herbal medicine [13]. Moreover, in folk medicine, women usually employed *S. rosmarinus* for menstrual complaints [14].

Most of the effects of *S. rosmarinus* are related to its phytochemical composition, which consists of various families of bioactive compounds such as polyphenols and phenolic terpenoids, besides its rich amounts of essential oil [2,15]. Its polyphenols include rosmarinic acid, while its phenolic terpenoids include carnosic acid, carnosol, ursolic acid as well as caffeic acid. These compounds lend the well-known beneficial effects to *S. rosmarinus* extract [16,17].

*S. rosmarinus* is broadly recognized as one of the species with the highest antioxidant activity [13]. Although this powerful antioxidant activity is due to the synergistic actions of several metabolites present in the plant, these has been attributed to its major polyphenol, rosmarinic acid, and to the two mains phenolic diterpenes, carnosol and carnosic acid [18,19]. Likewise, the presence of these three compounds is principally responsible for the anti-inflammatory property of rosemary belonging to its ursolic, micromeric and oleanolic acids [1,16].

The beneficial effects of *S. rosmarinus* in the prevention and treatment of skeletal muscle atrophy have been also recognized. Indeed, a recent study highlighted that carnosol was implicated in the decrease in skeletal muscle loss by reducing the ubiquitin–proteasome system-dependent protein degradation pathway, which inhibits the E3 ubiquitin ligase muscle ring finger protein-1 [20].

Furthermore, *S. rosmarinus* ameliorates the lipids profile; indeed, Soliman showed that *S. rosmarinus* improves lipid metabolism in a streptozotocin-induced diabetic model [21], while Wang et al. showed that an ethanol extract of *S. rosmarinus* significantly reduced the amounts of triglycerides, free fatty acids, and total cholesterol in the liver in an animal model of non-alcoholic fatty liver disease (NAFLD) [22]. Therefore, these findings suggest that the antihyperlipidemic effects of *S. rosmarinus* could have potential benefits in the prevention of NAFLD, improving hepatic function and avoiding the spread of complications related to this disease [23,24].

The phytochemical composition of the *S. rosmarinus* extracts is affected by the area of collection, environmental factors, part of the plant used and the harvest time [25]. Indeed, the phenolic diterpenes and rosmarinic acid contents as well as the flavones vary according to different geographical regions of growth [26]. It has been reported that the levels of these compounds are higher in the warm months [27]. Moreover, the chosen method of extraction and the parameters used affect the phytochemical composition and beneficial properties of the extract. Generally, extracts containing non-volatile compounds were obtained using conventional maceration or Soxhlet extraction [28]. Maceration is an extraction method based on solid–liquid separation, with the liquid phase consisting of an organic solvent, water or a mixture of organic solvent and water in which the solid phase is immersed [29]. However, maceration as well as Soxhlet extraction are affected by some disadvantages, including long extraction times, high solvent consumption and/or a degradation of thermolabile compounds [30]. Several studies reported using different extraction techniques that improve the efficiency of extraction from *S. rosmarinus*, minimizing the amounts of solvents and preventing the decomposition of natural bioactive compounds in the extracts [31]. The most investigated methods for obtaining rosemary extracts are ultrasound-assisted extraction (UAE), microwave-assisted extraction (MAE), pressurized liquid extraction (PLE), and supercritical fluid extraction (SFE) [32].

Among these methods, ultrasound sonication has been extensively exploited to improve the bioactive compounds’ extraction yield, since it is recognized as a faster and more efficient technique. Ultrasound sonication generally requires twenty minutes compared to the at least 12 h required for the maceration method [33]. Thus, in a shorter time, sound waves are used to enhance the recovery of polyphenolic compounds from the samples mixed with solvents in the flask through cell wall rupture [34]. Based on this evidence and to enhance the efficiency of extraction, a 3-h maceration protocol using ethanol/water followed by sonication were applied to obtain a higher total phenolic content [34,35]. Beyond the extraction method, the temperature and the type of the solvent play a key role in the recovery of the bioactive compounds of *S. rosmarinus*. It has been demonstrated that high processing temperatures could lead to the degradation of rosmarinic acid, carnosic acid and carnosol, since they are thermolabile compounds, which reduces the bioactivity of the obtained extracts [36]. Moreover, the use of ethanol as a solvent increases the rosmarinic acid content obtained, and ethanol coupled with sonication increases the total phenolic contents by more than three times [32]. Another evidence highlighted that the extraction yield of rosmarinic acid and terpenoids was higher when ultrasound was coupled with 70% or 90% ethanol [37].

The aim of our research was to compare the antioxidant and antihyperlipidemic effects of a terpenoids-rich *Salvia rosmarinus* extract (TR*Sr*E) and a polyphenols-rich *Salvia rosmarinus* extract (PR*Sr*E) obtained through a unique maceration process followed by ultrasound sonication extraction. After the phytochemical characterization of the two extracts was carried out with HPLC, the antioxidant power was evaluated using electron paramagnetic resonance (EPR), which is the only method that allows us to identify and directly quantify free radical species [38]. Particularly, the scavenging activity of a *S. rosmarinus* extracts against hydroxyl radicals was evaluated for the first time using this spectroscopy method. Then, the capability to reduce lipid accumulation was evaluated in McA-RH7777 cells exposed to oleic acid (OA) as an in vitro model of NAFLD.

## 2. Materials and Methods

### 2.1. Plant Material

The aerial parts of *S. rosmarinus* Spenn. (Figure 1) were harvested in June, in the geographical area of Trebisacce (39°51′24.6″ N 16°30′02.3″ E, Cosenza, Calabria). The taxonomic identification was confirmed by Dr. C. Lupia, Department of Health Sciences, University “Magna Graecia” of Catanzaro, and a specimen was preserved in the Mediterranean Ethnobotanical Conservatory (Sersale, Catanzaro, Italy) with the following accession number for *S. rosmarinus* Spenn.: Lamiaceae section, 73.

### 2.2. Extraction Procedure

The aerial parts of *S. rosmarinus* Spenn. were dried at 40 °C for 72 h in the dark, and then minced in liquid nitrogen with a mortar and pestle. The plant material was extracted with a mixture of ethanol/water (EtOH/w, 75/25, % *v*/*v*), through maceration for 3 h at room temperature (RT) in the dark, followed by sonication for 20 min. The aerial parts/solvent ratio was 1 g/20 mL. The sonication was performed with a frequency of 20 kHz, using an ultrasonic homogenizer (Sonopuls model HD 2070, Bandelin, Berlin, Germany) equipped with a titanium alloy flat-tip probe (13 mm diameter; VS 70 T, Bandelin, Berlin, Germany), in a cooling bath, to avoid the degradation of phenolic compounds due to the high temperatures. The sonication was employed at a controlled amplitude with an output power of 70%, and in a continuous mode of operation (10 cycles). Then, the solution was centrifuged at 4 °C and 1036 rcf for 5 min and filtered through filter paper. From the supernatant obtained, the solvent was removed via evaporation at 40 °C, and then the residue was completely dried to achieve a terpenoids-rich *Salvia rosmarinus* extract (TR*Sr*E). The aqueous phase was passed through a polystyrene-absorbing resin column (Mitsubishi Chemical Group, Tokyo, Japan) to concentrate the polyphenolic compounds [39] and to eliminate sugar impurities. The entrapped polyphenols were recovered with ethanol; then, following the evaporation of the solvent, a polyphenols-rich *Salvia rosmarinus* extract (PR*Sr*E) was obtained.

### 2.3. HPLC Analysis

The phytochemical characterization and quantification of the two *S. rosmarinus* dry crude extracts were carried out with HPLC through a PerkinElmer Flexar Module equipped with a series 200 autosampler, a series 200 Peltier LC column oven, a series 200 LC pump and an Agilent 4μ C18 100A (250 × 4.6 mm) column. The HPLC system was coupled to a photodiode array (PDA) detector, and HPLC analysis was performed with Chromera software (version 3.4.0.5712).

Totals of 61.7 mg of the dry TR*Sr*E and 20 mg of the dry PR*Sr*E were dissolved in 10.0 mL of EtOH (100%) and 10.0 mL of EtOH/w (50/50; % *v*/*v*), respectively, then vortexed until dissolution was complete; then, the samples were filtered with a 0.2 μm PTFE filter. Finally, 10 μL of each sample was injected into the HPLC system. A two-solvent gradient (0.88% trifluoroacetic acid/acetonitrile) was used for the elution with a flow of 0.7 mL/minute. The detection wavelength was set at 285 nm, while the column temperature was set at 30 °C.

### 2.4. DPPH Assay: Radical Scavenging Activity

The free radical scavenging potential was evaluated via a modified 1,1-diphenyl-2-picrylhydrazyl (DPPH) assay; a traditional method is generally performed to assess the scavenging activity of an extract.

In a methanolic solution of DPPH (40 mg/mL), 10 μL of six different concentrations of the *S. rosmarinus* Spenn. extracts were added (0, 0.25, 0.5, 1.0, 2.0 and 5.0 mg/mL). After 30 min at 25 °C in the dark, the absorbances of the resulting solutions were measured through a UV–Vis spectrophotometer (Multiskan GO, Thermo Scientific, Denver, CO, USA) at 517 nm. The results of the DPPH assay are expressed in terms of inhibition % and IC_50_ value (the concentration required to scavenge 50% of the DPPH radical). Ascorbic acid was used as a positive control, and all tests were carried out in triplicate.

### 2.5. Evaluation of Radicals Scavenging via Electron Paramagnetic Spectroscopy (EPR)

The scavenging activity of the *S. rosmarinus* Spenn. extracts against DPPH and hydroxyl (^•^OH) radicals was assessed using electron paramagnetic spectroscopy (EPR), as previously described [34].

The DPPH radical scavenging capacity of the *S. rosmarinus* extracts was evaluated, adding 50 μL of the tested extract (5 mg/mL concentrations of each) to 200 μL of methanolic DPPH solution (0.1 mM), mixing, and then acquiring the EPR spectra after 1 min of reaction. Ascorbic acid was used as a positive control.

Moreover, the capability of the *S. rosmarinus* Spenn. extracts to reduce hydroxyl radicals (OH^•^) was determined. Hydroxyl radicals (10^−9^ s half-life) were generated through a non-catalytic Fenton reaction, and BMPO (5-tert-butoxycarbonyl-5-methyl-1-pyrroline-N-oxide, B568-10, Dojindo EU GmbG, Munich, Germany) was used as a spin trap.

The EPR acquisitions were performed 1 min after mixing 15 μL of the BMPO solution (1.5 mg were dissolved in 5 mL of ddH_2_O), 75 μL of 1 mM H_2_O_2_, 75 μL of 100 μM iron (II) sulfate heptahydrate (FeSO_4_ • 7H_2_O, 7782-63-0, Merck KgaA, Darmstadt, Germany) and 50 μL of ddH_2_O. A volume of 75 μL of the corresponding *S. rosmarinus* Spenn. extract (5 mg/mL ethanolic solution) was added to the reaction mixture after the production of the hydroxyl radical. Ascorbic acid (5 mg/mL) was used as a positive control. The solutions of the single components were prepared the same day of the analysis.

All of the EPR spectra were acquired in the X band (9.43 GHz) using a Bruker Magnettech ESR5000 (Bruker Biospin MRI GmbH, Ettlingen, Germany) with the following experimental parameters: 0.05 mT modulation amplitude, 336.64 mT central field, 12.00 mT sweep, 30 s sweep time, a modulation frequency of 100 Khz, 8 accumulations, and 20 mW (for DPPH radical) or 6 mW microware power (for BMPO-OH spin adduct).

To assess the total amount of free radicals in each acquisition and evaluate the radical scavenging activity of the *S. rosmarinus* Spenn extract, the spectral areas were integrated and calculated (OriginPro 2018). Finally, the scavenger percentage was quantified using the following formula, as described by Lamponi et al. (2021) [40]:scavenger % = (A_0_ − A_extract_/A_0_) × 100

### 2.6. Total Phenolic Content (TPC) Determination

The total phenolic content (TPC) of the *S. rosmarinus* extracts was determined according to the spectrophotometric Folin–Ciocalteau modified method [41]. A 100 μL aliquot of stock extract solution was mixed with 500 μL of 2 N Folin–Ciocalteu’s phenol reagent (5 min of incubation) and 400 μL of 10.75% *w*/*v* anhydrous sodium carbonate (*w*/*v*) (incubation for 25 min). The absorbance was measured at 760 nm in a UV–Vis spectrophotometer (Multiskan GO, Thermo Scientific, Denver, CO, USA). The calibration curve was carried out using gallic acid as the standard solution (y = 0.0033x; R^2^ = 0.9966), and the results of the TPC are expressed as mg of gallic acid equivalent per gram of dry weight (mg GAE/g dw).

### 2.7. Total Flavonoid Content (TFC) Determination

The total flavonoid content (TFC) of the *S. rosmarinus* extracts was quantitated according to the aluminum chloride method, using rutin as the standard. Following 5 min of incubation at RT of 100 μL of the stock extract solution (200 μg/mL) with 30 μL of NaNO_2_ 5%, the solution was mixed with 30 μL of 10% AlCl_3_ (incubated for 5 min at RT). Afterward, 200 μL of NaOH (1M) was added to the solution that was incubated for 5 min at RT. The mixture was vortexed, then incubated for 10 min at RT. Finally, the absorbance was measured at 513 nm in a UV–Vis spectrophotometer (Multiskan GO, Thermo Scientific, Denver, CO, USA). The calibration curve was calculated using rutin as the standard solution (y = 0.0008x; R^2^ = 0.9973), and the results of the TFC are expressed as mg of rutin equivalent per gram of dry weight (mg RE/g dw).

### 2.8. Cell Culture

McA-RH7777 is a rat Morris hepatoma-derived cell line obtained from the American Type Culture Collection (ATCC), Rockville, MD ~CRL 1601. The cells were cultured in Dulbecco’s modified Eagle’s high glucose medium (DMEM W/Glutamax-I, Pyr,4.5 g Glu-31966047-Gibco, Waltham, MA, USA) at 37 °C under 5% CO_2_ in a humidified 95% atmosphere. The DMEM was supplemented with 10% *v*/*v* fetal bovine serum (FBS, Qualified, Hi,10500064-Gibco, Waltham, MA, USA), 10,000 U/mL penicillin, and 10 mg/mL streptomycin (Penicillin Streptomycin Sol, 15140122-Gibco, Waltham, MA, USA).

### 2.9. Free Fatty Acid (FFA) Exposure and S. rosmarinus Spenn. Treatment

The cells were cultured in Dulbecco’s modified Eagle’s high glucose medium (DMEM) at 37 °C under 5% CO_2_ in a humidified 95% atmosphere. The DMEM was supplemented with 1% FBS, 10,000 U/mL penicillin and 10 mg/mL streptomycin (Penicillin Streptomycin Sol, 15140122-Gibco).

To reproduce an in vitro model of NAFLD, oleic acid (OA) was chosen. Free fatty acid (FFA) was dissolved in 100% ethanol to prepare the stock solution of OA 240 mM (O1008-Sigma-Aldrich). The cells were exposed to exogenous FFA (100 μM) for 24 h. Moreover, the FFA was complexed with bovine serum albumin (BSA) 33.3 μM at a 3:1 molar ratio. The two *S. rosmarinus* Spenn. extracts were dissolved in DMSO to reach a stock concentration of 80 mg/mL. Increasing concentrations of *S. rosmarinus* extracts (25 μg/mL, 50 μg/mL, 100 μg/mL and 150 μg/mL) were used to treat the cells exposed to OA.

### 2.10. MTT Assay

An MTT colorimetric assay was performed to evaluate cell viability. This colorimetric assay is based on the reduction of a yellow tetrazolium salt to a water insoluble blue formazan crystal by dehydrogenases of metabolically active cells [42]. Using 1 × 10^4^ cells/well plated in 96 well dish, the exposure was for 24 h to OA 100 μM and the two extracts of *S. rosmarinus* at increasing concentrations (25 μg/mL, 50 μg/mL, 100 μg/mL and 200 μg/mL). Subsequently, 0.5 mg/mL of the MTT was added to the medium. Then, after 4 h of incubation at 37 °C, the generated blue formazan crystals were dissolved in DMSO. Finally, the resulting-colored solution was quantified by measuring the absorbance at 570 nm and 690 nm (Blank) through a microplate reader (Multiskan GO, Thermo Scientific, Denver, CO, USA). The absorbance of the untreated controls was taken as 100% survival.

### 2.11. Determination of Intracellular Total Fatty Acid Content

The total intracellular fatty acid accumulation was determined using Oil Red O and Nile red cell stainings [43].

In regard to the Oil Red O cell staining, 5 × 10^4^ cells/well were treated with FFA and the two *S. rosmarinus* extracts at concentrations of 25 μg/mL and 50 μg/mL for 24 h. Afterward, the cells were washed twice using PBS and fixed with 3% paraformaldehyde for 15 min. Finally, Oil Red O staining at 3.3 μg/mL for 8 min (ORO 1.02419-Sigma-Aldrich) was used to stain the intracellular lipids, whereas DAPI (D8417, Sigma Aldrich, Milan, Italy) was used to stain the cell nuclei.

For Nile red staining, 1 × 10^4^ cells/well plated in a 96-well dish were exposed for 24 h to OA 100 μM and increasing concentrations of the two *S. rosmarinus* extracts (25 μg/mL, 50 μg/mL and 100 μg/mL). Afterward, following the medium’s removal, the cells were washed with PBS, then incubated with 0.75 μg/mL AdipoRed^TM^ Reagent (PT-7009, Lonza, Basel, Switzerland) dye for 15 min at RT. The Nile red fluorescence was determined using a Fluoroskan Ascent Microplate Fluorometer (Thermo FisherScientific, Waltham, MA, USA) with 485 nm excitation and 535 nm emission.

### 2.12. Fluorescence Image Acquisition

The acquisition of ORO-DAPI stained fluorescence images was performed using a confocal microscope TCS SP5 (Leica Microsystems, Wetzlar, Germany) with a 63X objective. The lipid droplets analysis was carried out using ImageJ Fiji (version 2.3.0/1.53f), and the positive pixels percentages were used as measures of the total fatty acid accumulation. At least six images for each group were acquired, and then processed using a minimum threshold value of 90–255, which was selected and kept constant during all of the analyses.

### 2.13. Statistical Analysis

All of the statistical analyses were performed using GraphPad PRISM 9.3.1 (GraphPad Software, Inc., La Jolla, CA, USA). All the results are shown as mean ± S.E.M. The normality was tested using the Shapiro–Wilk normality test. The data were analyzed via one-way ANOVA followed by Tukey’s test (for normally distributed data), or by a Kruskal–Wallis analysis of variance followed by Dunn’s tests (data without normal distribution). Comparisons of the data derived from the two groups were performed with the unpaired two-tailed Student’s *t* test or the Mann-Whitney test. Values with *p* < 0.05 were considered statistically significant. The DPPH assay data were fitted using nonlinear regression to compute the IC_50_ values. A correlation analysis between the TPC/TFC and antioxidant activity (DPPH assay, DPPH-EPR, BMPO-^•^OH-EPR) was carried out using Pearson’s correlation coefficient.

## 3. Results

### 3.1. Phytochemical Characterization and Quantification via HPLC

The phytochemical characterization of the two *S. rosmarinus* dry extracts performed through HPLC showed the presence of rosmarinic acid among the polyphenols, and the presence of carnosic acid and carnosol among the terpenoids.

As shown by the chromatograms, PR*Sr*E was abundant in rosmarinic acid (19%) and had a lower concentration in terpenoids (0.6% carnosic acid and 0.3% carnosol, Figure 2A), while the major component in TR*Sr*E was carnosol (7.2%) followed by rosmarinic acid (2%) and carnosic acid (1.9%, Figure 2B).

### 3.2. In Vitro Antioxidant Activity of S. rosmarinus Spenn. Extracts

First, the antioxidant capacities of the PR*Sr*E and TR*Sr*E were assessed using the classical DPPH assay. The strong relation between concentration and percentage of inhibition was explained with a nonlinear regression. As shown in Figure 3, both extracts showed a concentration-dependent radical scavenging activity. The PR*Sr*E exerted a higher free radical (DPPH) scavenging activity (IC_50_: 2.57 ± 0.22 mg/mL) compared to the TR*Sr*E (IC_50_: 3.27 ± 0.14 mg/mL). As a reference, the IC_50_ value of ascorbic acid was 1.85 ± 0.22 mg/mL.

### 3.3. DPPH and Hydroxyl Radicals Scavenging Activity of S. rosmarinus Spenn. Extracts through EPR Spectroscopy

The scavenging activities of the *S. rosmarinus* Spenn. extracts against DPPH and hydroxyl (^•^OH) radicals were also evaluated. The decrease in the EPR signal intensity was related to the radical scavenging activity of the extracts.

As shown in Figure 4, the DPPH-EPR spectrum highlighted a six-line pattern with an integrated spectral area value of 614.52 a.u. Although both *S. rosmarinus* Spenn. extracts showed DPPH radical scavenging capacity, the PR*Sr*E showed a higher antioxidant capacity with a scavenger percentage equal to 93.11% (∫ = 42.35 a.u., Figure 4) compared to the TR*Sr*E (∫ = 53.05 a.u., scavenger percentage 91.37%, Figure 4). As the positive control, the integrated spectral area value and the scavenger percentage of ascorbic acid were 46.25 a.u. and 92.47%, respectively (Figure 4).

In regard to the evaluation of the scavenging activity against hydroxyl radical (^•^OH), the EPR showed the typical 4-line spectrum of a BMPO-^•^OH adduct with an integrated spectral area value of 43.05 a.u.

A higher scavenging activity against hydroxyl radicals with a scavenger percentage of 65.23% was exerted by the PR*Sr*E (∫ = 14.97 a.u., Figure 5) compared to the TR*Sr*E (∫ = 22.06 a.u., scavenger percentage 48.76%, Figure 5). Ascorbic acid was used as the positive control (∫ = 21.94 a.u., 49.04%, Figure 5).

### 3.4. Characterization of Total Phenolic and Flavonoid Contents of S. rosmarinus Extracts

The phytochemical compounds analysis showed that the total phenolic content was significantly higher in the PR*Sr*E (255.6 ± 10.98 mg GAE/g dw) than in the TR*Sr*E (42.80 ± 4.42 mg GAE/g dw, *p* < 0.001, Figure 6). Likewise, the total flavonoid content of the PR*Sr*E was significantly higher (1065 ± 18.83 mg RE/g dw) compared to the TR*Sr*E (208.4 ± 16.33 mg RE/g dw, *p* < 0.001, Figure 6).

### 3.5. Antioxidant Activity Is Significantly Correlated with Total Phenolic and Flavonoid Contents

The Pearson’s correlation analyses showed that both the TPC and TFC were related with antioxidant activity (DPPH assay, DPPH-EPR, BMPO-^•^OH-EPR). Indeed, as shown in Table 1, a strong correlation was found between the antioxidant activity and TPC (r: −0.975, *p* < 0.001) and TFC as well (r: −0.993, *p* < 0.001). Therefore, the higher antioxidant activity exerted by the PR*Sr*E compared to the TR*Sr*E was due to the higher TPC and TFC in the aqueous extract (Figure 7).

### 3.6. Treatment with Terpenoids-Rich or Polyphenols-Rich Salvia rosmarinus Extracts Does Not Affect Cell Viability

The screening of the viability of the McA-RH7777 in response to its exposure to OA 100 μM, vehicle (DMSO) or treatment with the *S. rosmarinus* extracts was performed through an MTT assay [44].

The viability of the hepatoma cells was not reduced following exposure to OA 100 μM as well as after the treatment with increasing concentrations of PR*Sr*E or TR*Sr*E (Figure 8).

The exposure to DMSO led to a significant, dose-dependent decrease in the viability of the McA-RH7777 cells, with a maximum decrease at a % of vehicle equivalent to a concentration of 200 μg/mL (Figure 8).

### 3.7. Salvia rosmarinus Spenn. Extracts Reduce Intracellular Lipid Accumulation

The Oil Red O (ORO) and Nile Red stainings showed that the 24h exposure to OA 100 μM led to a significant accumulation of cytoplasmic lipid droplets in the McA-RH7777 (*p* < 0.001, Figure 9 and Figure 10). The treatment with PR*Sr*E or TR*Sr*E significantly reduced the intracellular lipid accumulation at a concentration of 50 μg/mL (Figure 9 and Figure 10). The ORO staining highlighted a reduction in intracellular lipid accumulation at a concentration of 25 μg/mL as well, with both extract treatments (*p* < 0.001, Figure 9). Furthermore, the TR*Sr*E exerted a better anti-lipidemic activity at 50 μg/mL compared to the PR*Sr*E (Figure 9 and Figure 10).

## 4. Discussion

The bioactivity of plant extracts depends on their phytochemical profiles, which may be influenced by several factors, such as the harvesting period, geographical origin, chemical properties of the compounds, the extraction, and the purification methods used as well as the chosen solvent. The polarity of the latter covers an important role in the selective extraction of natural antioxidants [45].

The extraction of carnosic acid, carnosol and rosmarinic acids, which are known as the main antioxidants in *S. rosmarinus*, is mainly carried out in acetone, pure ethanol or hydroalcoholic mixtures [46]. Noteworthily, a mixture ranging from 50% to 80% ethanol results in the highest extraction yields and total bioactive compounds recoveries in comparison to lower EtOH or pure solvents levels [47].

Furthermore, ultrasound is a useful technology, as it does not require complex instruments and is relatively low-cost, ensuring a higher extractive value compared to traditional extraction methods such as Soxhlet, which may have some drawbacks such as their high temperature and long processing times that could affect thermolabile and unstable compounds, as well as their low selectivity and elimination of solvent residues that are often prohibited by international food and cosmetic regulations [48]. Moreover, the use of ultrasound along with ethanol improves the extraction performance of antioxidant compounds compared with other solvents (i.e., methanol) when dried plant material is used [49]. In this study, a 3-h maceration with 75% EtOH followed by sonication of dried leaves of *S. rosmarinus* was performed. To purify the hydroalcoholic extract and separate molecules as a function of their physicochemical properties, we first removed EtOH via a rotary evaporator [28]. The decrease in EtOH led more lipophilic molecules to precipitate in residual water, which contained hydrophilic compounds, such as polyphenols, flavonoids and short-chain organic acids [50]. The presence of physically different components of the mixture allowed us to separate the solid and liquid phases obtained from the extract via a mechanical filtration procedure. Therefore, a precipitate mainly containing terpenoids and aromatics polycyclic, named TR*Sr*E, was mechanically separated from the hydrophilic phase via filtration through a cellulose filter. The aqueous phase was finally purified using a chromatographic column selective for anions, and an extract concentrate in phenolic compounds [51], named PR*Sr*E, was obtained. This strategy allowed us to obtain 96 mg of TR*Sr*E and 35 mg of PR*Sr*E of dry extracts. Each gram of rosemary underwent extraction, corresponding to a total percentage yield of 13.1%. This percentage was in line with data reported by Lamponi et al., from another Italian rosemary collected in Tuscany [40], but our yield was different from the data published by Jacotet-Navarro et al. or by Nguyen-Kim et al., which reported extraction yields of around 25% and 33%, respectively, with 75% EtOH for *S. rosmarinus* collected in Morocco [47] and in Vietnam, respectively [52]. Since these authors used a conventional method of extraction, this difference may be ascribed to the environmental conditions, as the chemical composition of rosemary can be affected by soil properties [53], fertilizers [54], saltiness [55], humidity and temperature [56]. Furthermore, Balouiri et al., using a 22-h two-step maceration with hexane and methanol, were able to achieve an extraction yield of 15.8%, in line with our data, whereas, using an ultrasonic extraction, a yield of 8.7% was obtained [52].

The antioxidant power of a certain extract is generally due to its content in polyphenol and flavonoid compounds [51]. In our study, the TPC for the PR*Sr*E was 255.6 ± 10.98 mg GAE/g dw, whereas in the TR*Sr*E the TPC was 42.80 ± 4.42 mg GAE/g dw. Pontillo et al., using an enzyme-assisted extraction as a pre-treatment for maceration, achieved a maximum TPC of 15.2± 0.3 mg GAE/g, whereas a microwave-assisted extraction with various ratios of hydroalcoholic solution obtained a TPC in a range between 3 ± 0.3 and 8.9 ± 06 mg GAE/g [57]. Furthermore, in a recent study, it was shown that using a Soxhlet extraction, a TPC range between 15 ± 1.93 mg GAE/g and 34.72 ± 1.65 mg GAE/g, depending on the solvent used, could be obtained in a wild rosemary from northern Morocco [58].

Regarding the flavonoid content, the PR*Sr*E showed a TFC of 1065 ± 18.83 mg RE/g dw, and the TR*Sr*E showed a TFC of 208.4 ± 16.33 RE/g dw. Butu et al. achieved TFC values of 337.97 ± 0.50 mg RE/g and 243.63 ± 0.17 mg RE/g via Soxhlet and percolation methods, respectively [59].

Therefore, our TPC and TFC results are coherent with the values reported in the literature in regard to the TR*Sr*E, whereas the higher TPC and TFC values showed by the PR*Sr*E were due to the adopted strategy to concentrate polyphenolic compounds through the polystyrene absorbing resin column.

Overall, the optimization of our protocol of ultrasound-assisted maceration also offered a valid tool to obtain a more powerful extraction selectivity compared to a simple maceration technique. Indeed, the phytochemical analysis of the PR*Sr*E showed a higher concentration of rosmarinic acid (19%) versus carnosic acid (0.6%) and carnosol (0.3%), whereas the TR*Sr*E exhibited an enrichment in carnosol (7.2%) and carnosic acid (1.9%) followed by rosmarinic acid (2%).

Based on these results and on the biological properties referred to *S. rosmarinus* [60], we tested the effects of PR*Sr*E and TR*Sr*E on the hepatoma McA-RH7777 cell line and, specifically, their antioxidant and antilipidemic activities.

To verify the antioxidant properties, we first performed the classical DPPH assay [61] that highlighted the concentration-dependent radical scavenging activity of both studied extracts, although the PR*Sr*E exerted a higher free radical scavenging power than the TR*Sr*E. A further characterization, carried out using EPR, demonstrated that this difference can be attributed to the diversified scavenging activity against hydroxyl radical (^•^OH) detected through the typical 4-line spectrum of the BMPO-^•^OH adduct. The analysis demonstrated that the PR*Sr*E displayed a hydroxyl radical scavenging percentage of 65.23%, compared to the 48.76% of the TR*Sr*E. On the other hand, the TR*Sr*E exerted a better anti-lipidemic activity compared to the PR*Sr*E at 50 μg/mL.

This latter result, highlighting the different activities of the two extracts, may underline the added value of TR*Sr*E in counteracting the development of metabolic diseases such as NAFLD/MAFLD [62,63]. The first step in NAFLD is represented by lipid accumulation in hepatocytes, triggering free radical overproduction that further amplifies cell damage [23]. Evidence exists that the biological activity of carnosic acid can be due to its anti-obesity properties. Indeed, it can regulate fatty acid metabolism and can inversely control the expression of hepatic lipogenesis-related genes (L-FABP, SCD1 and FAS) and of a lipolysis-related gene (CPT1) [64]. In addition, carnosol and carnosic acid revealed hypoglycemic and hypolipidemic effects by the activation of signaling pathways, including AMP-activated protein kinase (AMPK) and PPAR-γ, and by the up-regulation of the low-density lipoprotein cholesterol (LDL-C) receptor and PGC1α [65,66]. Thus, despite the lowest scavenging activity of TR*Sr*E against ^•^OH, this action, combined with the modulation of lipid metabolism and the inhibition of lipid accumulation, represents an added value. Indeed, it suggests a potential use of our extract in the prevention of hepatic steatosis, but also of heart dysfunction [67,68,69,70], confirming the cardioprotective properties of polyphenols and terpenoids [71,72].

Overall, these results demonstrated that the optimization of selective extraction methods of bioactive compounds from plants constitutes a fundamental approach to studying their biological activity that is aimed at supporting their use in several research fields.

## Figures and Tables

**Figure 1 plants-12-03306-f001:**
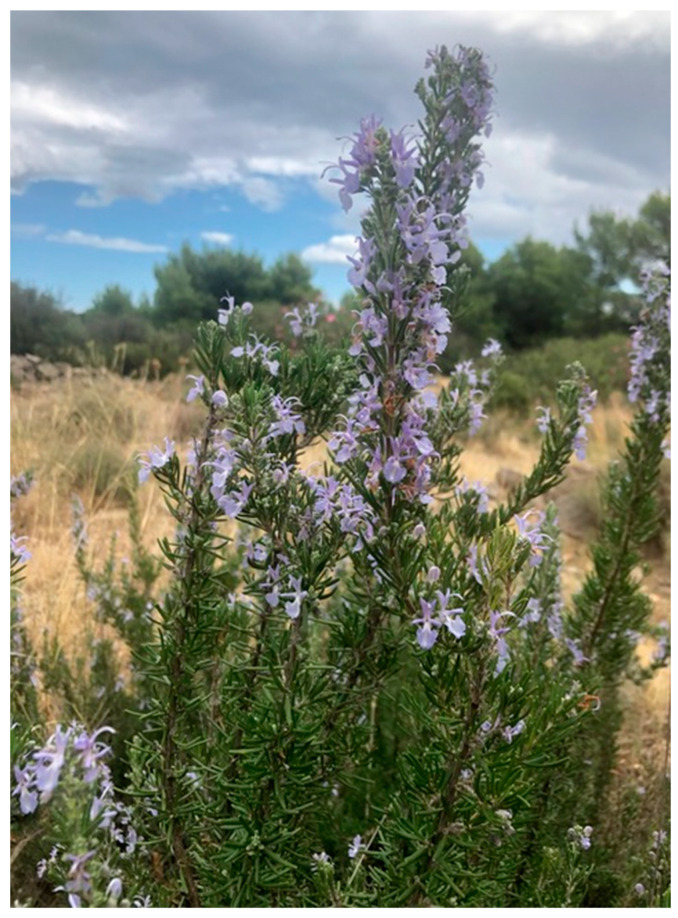
*S. rosmarinus* Spenn.

**Figure 2 plants-12-03306-f002:**
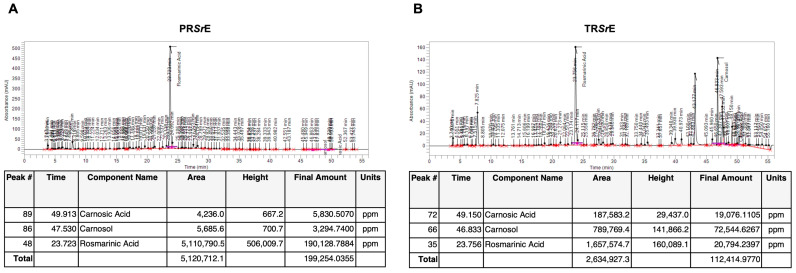
Phytochemical characterization and quantification of *S. rosmarinus* extracts. HPLC chromatograms of (**A**) PR*Sr*E and (**B**) TR*Sr*E.

**Figure 3 plants-12-03306-f003:**
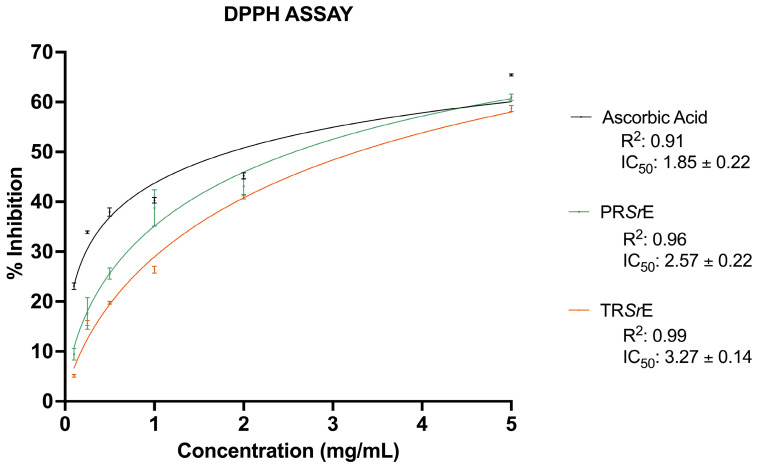
DPPH Assay. Antiradical scavenging activity expressed as % inhibition and IC_50_ values of PR*Sr*E and TR*Sr*E. Ascorbic Acid was used as positive control. Data were fitted with nonlinear regression to quantify the IC_50_ values. The results are expressed as mean ± SEM (*n* = 3).

**Figure 4 plants-12-03306-f004:**
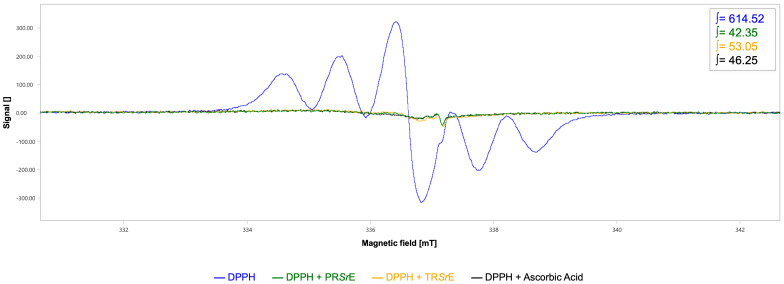
EPR spectroscopy for DPPH radical scavenging activity. EPR spectra and respective integrated spectral areas (∫) of DPPH in the absence (blue) and presence of PR*Sr*E (green) or TR*Sr*E (orange). Ascorbic acid was used as positive control (black).

**Figure 5 plants-12-03306-f005:**
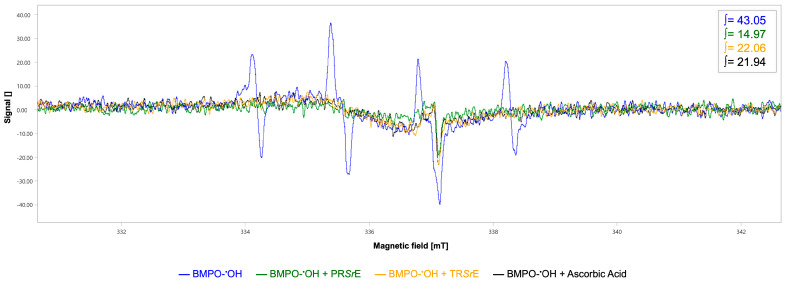
EPR spectroscopy for hydroxyl radical scavenging activity. EPR spectra and respective integrated spectral areas (∫) of BMPO-^•^OH adduct in the absence (blue) and presence of PR*Sr*E (green) or TR*Sr*E (orange). Ascorbic acid was used as positive control (black).

**Figure 6 plants-12-03306-f006:**
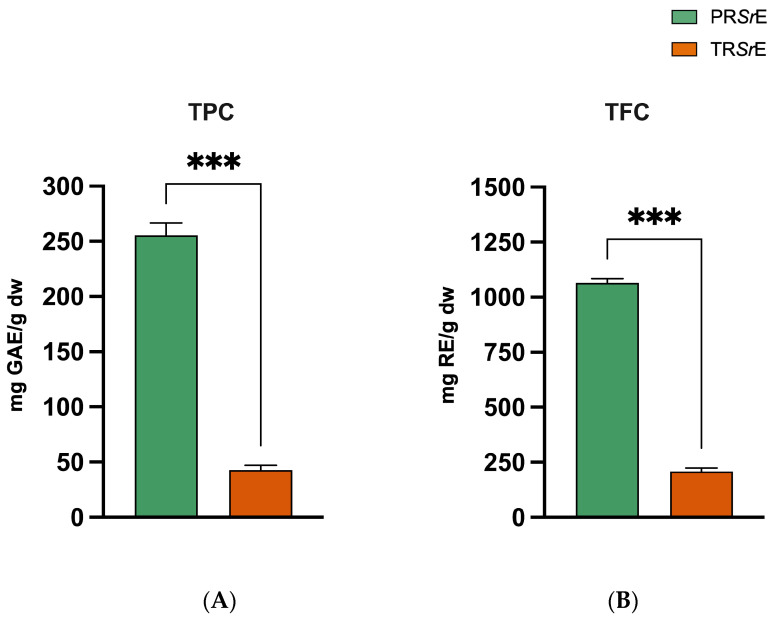
Phytochemical compounds analysis. (**A**) Total phenolic content (TPC) and (**B**) total flavonoid content (TFC) of PR*Sr*E and TR*Sr*E. The results are expressed as mean ± SEM. ***: *p* < 0.001.

**Figure 7 plants-12-03306-f007:**
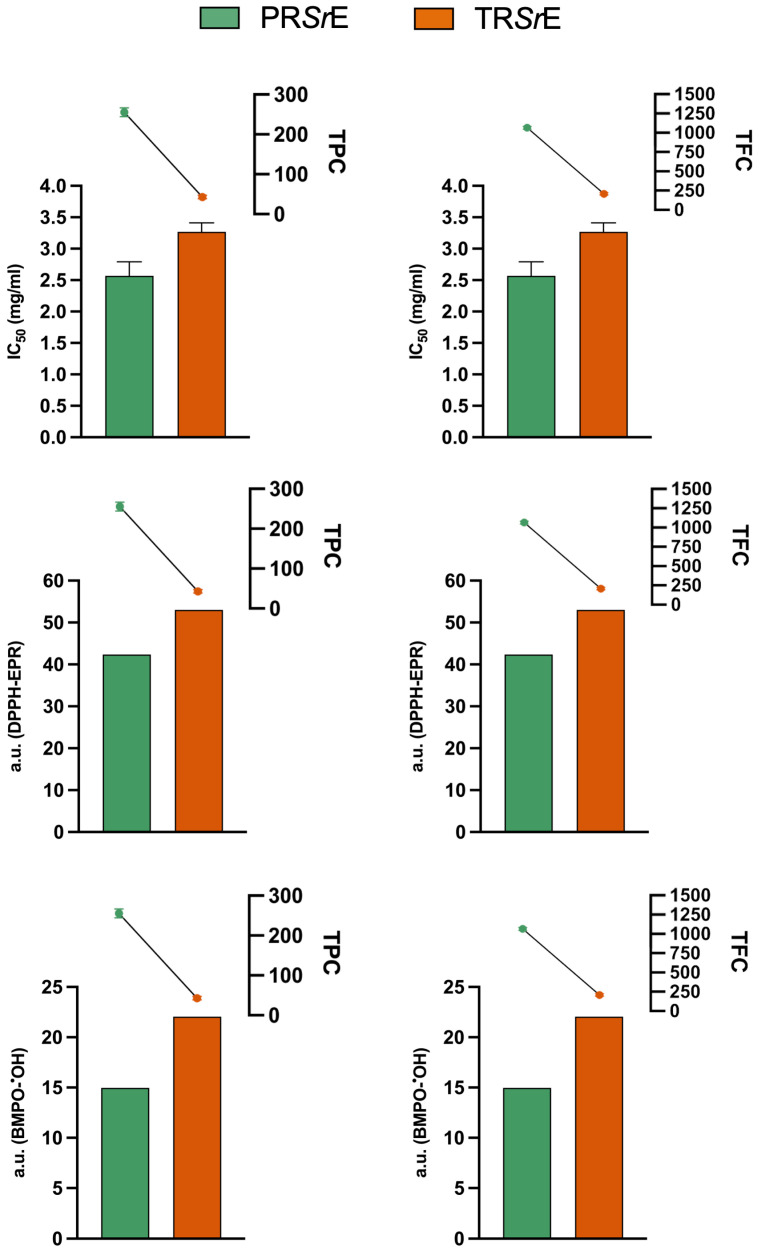
Correlation analysis.

**Figure 8 plants-12-03306-f008:**
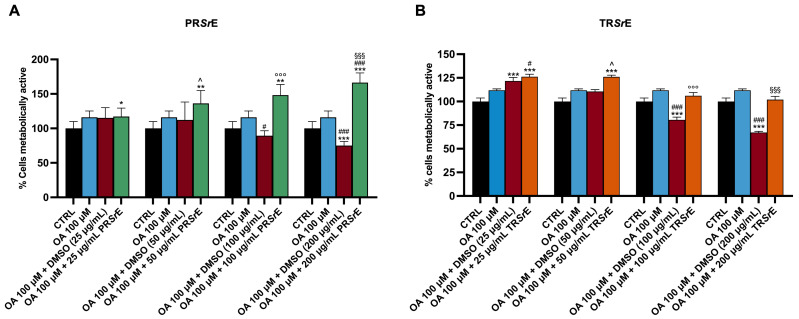
MTT assay for cell viability. Cell viability in response to exposure to OA 100 μM, vehicle (DMSO) or treatment with increasing concentrations of (**A**) PR*Sr*E or (**B**) TR*Sr*E. The results are expressed as mean ± SEM; *: *p* < 0.05, **: *p* < 0.01, ***: *p* < 0.001 vs. CTRl; #: *p* < 0.05, ###: *p* < 0.001 vs. OA 100 μM; ^: *p* < 0.05 vs. OA 100 + DMSO (50 μg/mL); °°°: *p* < 0.001 vs. OA 100 + DMSO (100 μg/mL); §§§: *p* < 0.001 vs. OA 100 + DMSO (200 μg/mL).

**Figure 9 plants-12-03306-f009:**
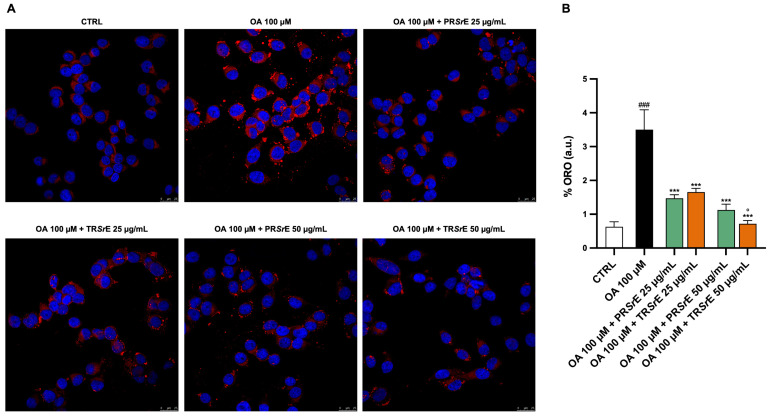
Total fatty acid accumulation assessed using Oil Red O (ORO) staining. (**A**) Representative confocal images of OA-induced lipid accumulation in McA-RH7777. (**B**) The results are expressed as mean ± S.E.M.; ###: *p* < 0.001 vs. CTRL; ***: *p* < 0.001 vs. OA 100 µM; °: *p* < 0.05 vs. OA 100 µM + PR*Sr*E 50 µg/mL.

**Figure 10 plants-12-03306-f010:**
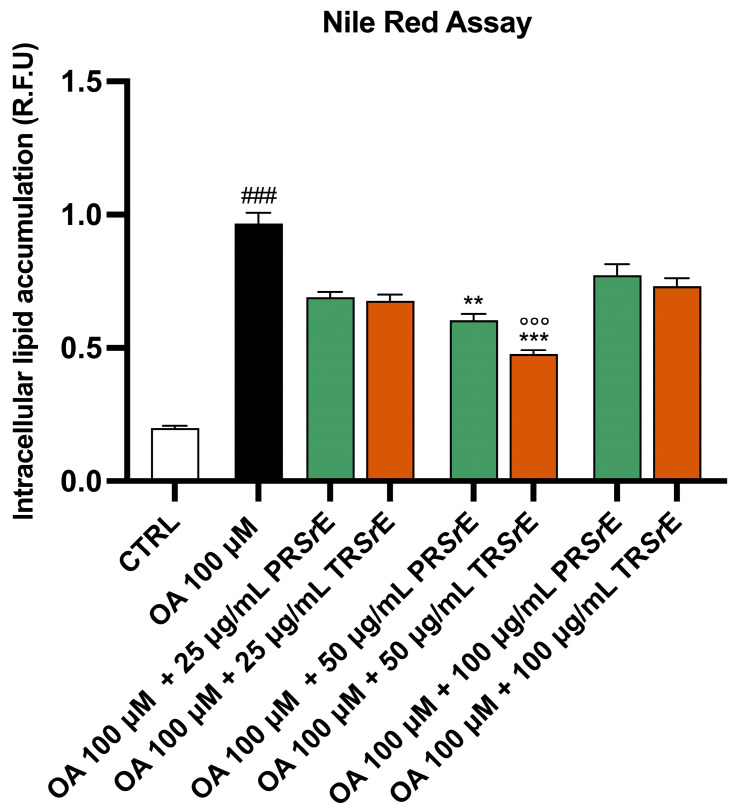
Total fatty acid accumulation assessed via Nile Red assay. Intracellular lipid accumulation in McA-RH7777. The results are expressed as mean ± S.E.M. ###: *p* < 0.001 vs. CTRL; **: *p* < 0.01; ***: *p* < 0.001 vs. OA 100 µM; °°°: *p* < 0.01 vs. OA 100 µM + PR*Sr*E µg/mL.

**Table 1 plants-12-03306-t001:** Correlation analysis.

	**Antioxidant Activity**	**Pearson’s r**	** *p* **
TPC	IC_50_, DPPH-EPR, BMPO-^•^OH	−0.975	<0.001
TFC	IC_50_, DPPH-EPR, BMPO-^•^OH	−0.993	<0.001

## Data Availability

The data presented in this study are available in the article. The original files are available on request from the corresponding author.

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
