# Peer review of "Salvia rosmarinus Spenn. (Lamiaceae) Hydroalcoholic Extract: Phytochemical Analysis, Antioxidant Activity and In Vitro Evaluation of Fatty Acid Accumulation"

_plants, 2023, doi:10.3390/plants12183306_

Round 1

Reviewer 1 Report

In the "Extraction procedure", indicate the mass of dry plant used, volume of the solvents.

In the discussion, compare the extraction yield with the standard methods used by others.

Line 208, add brackets at the beginning of the equation and before the x100

Enhance the quality of Fig.2 including its table.

Add standard error bars to the values (dots) in Fig.3

Enhance the quality of Fig.4 and 5 including the axis.

Add the name of Y axis in Fig. 10 and the unites.

In the discussion elaborate in the total phenols and flavonoids yield in this study and compare it with other studies used alternative methods of extraction.

Author Response

In the "Extraction procedure", indicate the mass of dry plant used, volume of the solvents.

R.1 First of all, we would like to thank the reviewer for the constructive criticism and time spent to analyze deeply this manuscript. We have now indicated the mass of dry plant used and volume of the solvents as required. The reviewer can find these in bold red in the Extraction procedure section.

In the discussion, compare the extraction yield with the standard methods used by others.

R.2 We would like to thank the reviewer for allowing us to better clarify this aspect. We have added in the discussion section further comparisons of the extraction yield achieved by others through standard methods.The reviewer can find these in bold red in the Discussion section.

Line 208, add brackets at the beginning of the equation and before the x100

R.3 We thank the reviewer for the suggestion. We have added the brackets as suggested.

Enhance the quality of Fig.2 including its table.

R.4 We thank the reviewer for the suggestion. We have enhanced the quality of Fig.2.

Add standard error bars to the values (dots) in Fig.3

R.5 We thank the reviewer for the suggestion. We have added the standard error bars to the values in Fig.3.

Enhance the quality of Fig.4 and 5 including the axis. 

R.6 We thank the reviewer for the suggestion. We have enhanced the quality of Figures 4 and 5.

Add the name of Y axis in Fig. 10 and the unites.

R.7 We thank the reviewer for the suggestion. We have added the name of the Y axis and the unites in Fig.10.

In the discussion elaborate in the total phenols and flavonoids yield in this study and compare it with other studies used alternative methods of extraction.

R.8 We would like to thank the reviewer this criticism. We have now discussed the TPC and TFC results as required. The reviewer can find these in bold red in the Discussion section.

Reviewer 2 Report

line 46 delete parentheses (and vinegar) change as ....and vinegar....

lines 71-72 change Muscle RING Finger  as ....muscle ring finger....

line 133 include voucher number 

2.1. Plant material 

Include GPS coordinates

 All manuscripts check and change S. Rosmarinus as  ..... S. rosmarinus 

lines 183, 187, 194, 204, .........422

line 215 change Phenolic Content as phenolic content

Line 225 change Flavonoid Content as flavonoid content

Minor editing of English

Author Response

line 46 delete parentheses (and vinegar) change as ....and vinegar....

lines 71-72 change Muscle RING Finger as ....muscle ring finger....

line 133 include voucher number 

2.1. Plant material 

Include GPS coordinates

All manuscripts check and change S. Rosmarinus as  ..... S. rosmarinus 

lines 183, 187, 194, 204, .........422

line 215 change Phenolic Content as phenolic content

Line 225 change Flavonoid Content as flavonoid content

  1. We thank the reviewer for the suggestion. Now we have made all the requested changes, and the reviewer can find these in bold red in the manuscript.